

# Application of problem-based learning and case-based learning integrated method in the teaching of maxillary sinus floor augmentation in implant dentistry

Yunfei Liu, Yamei Xu, Yueheng Li and Qingqing Wu

Stomatological Hospital of Chongqing Medical University; Chongqing Key Laboratory of Oral Diseases and Biomedical Sciences, Chongqing Municipal Key Laboratory of Oral Biomedical Engineering of Higher Education, Chongqing, China

## ABSTRACT

**Background**. Teaching of maxillary sinus floor augmentation (MSFA) is challenging for dental educators due to the varied sinus anatomy and high rate of complications. The method integrating problem-based learning and case-based learning (PBL-CBL method) may be more effective than the traditional teacher-centered method. The aim is to evaluate the efficacy of the PBL-CBL method in teaching MSFA.

**Materials & Methods**. Ninety-two students who received training between 2015 and 2017 at the Department of Implant Dentistry were divided randomly into an experimental group and a control group. Students in the experimental group were trained using the PBL-CBL method, while those in the control group were trained using the traditional teacher-centered method. After three months of training, a survey of the students' opinions about the corresponding teaching method was carried out through a feedback questionnaire. A theory test was used to investigate the level of MSFA knowledge among the students. A case analysis was designed to test whether the students can apply the knowledge in solving new problems.

**Results**. Compared with the control method, the PBL-CBL method resulted in higher scores in both the theory test and the case analysis, and obtained a higher rate of satisfaction among the students. The difference in scores between the two methods were statistically significant ($P < 0.01$),

**Conclusion**. The PBL-CBL method resulted in better results regarding acquisition of academic knowledge, ability in case analysis and student satisfaction compared with the teacher-centered method. It may be a promising mode for teaching complex surgical techniques in implant dentistry and other dental fields.

Corresponding authors
Yueheng Li,
YF@hospital.cqmu.edu.cn,
275772766@qq.com
Qingqing Wu,
501190@hospital.cqmu.edu.cn

## INTRODUCTION

Dental implants are widely applied for rehabilitation of partial and complete edentulism (*Pjetursson et al., 2012*; *Zitzmann et al., 2013*; *Fillion et al., 2013*). As an essential part of dental education, the teaching of implant dentistry has been required by multiple academic institutions (*De Bruyn et al., 2009*; *Stanford, 2005*). Pneumatization of the maxillary sinus

and atrophy of the alveolar ridge are common scenarios following the loss of posterior maxillary teeth. To develop these sites for dental implant placement,maxillary sinus floor augmentation (MSFA) are routinely performed. However, teaching MSFA faces great challenges. First of all, anatomy of maxillary sinus is highly varied, such as the aberrations of the maxillary septum and the different pathological conditions of Schneiderian membrane (*Malkinson & Irinakis, 2009*; *Irinakis, Dabuleanu & Aldahlawi, 2017*). The prevalence of maxillary septum is between 16% and 48% (*Naitoh et al., 2009*; *Rosano et al., 2010*; *Güncü et al., 2011*). The occurrence of Schneiderian membrane perforation is 10–60% of all procedures (*Becker et al., 2008*; *Nolan, Freeman & Kraut, 2014*). In addition, MSFA is technically sensitive because the surgical access to the sinus floor is quite limited, making teaching and training difficult. Many educators in the field of implant dentistry are working hard to find a suitable teaching method for MSFA so as to increase the teaching efficacy.

Given that most students learning MSFA are resident doctors, the teaching method of MSFA should take the characteristics of adult learning into consideration. Hallmarks of adult learning are the use of authentic problems to guide small-group discussions (*Abela, 2009*) and learning techniques facilitating retention of interest in the subject (*Major & Palmer, 2001*). The traditional teacher-centered teaching approach delivers basic and clinical sciences information primarily in a lecture format. Students learning in this way tend to rely on repetition and memorization (*Major & Palmer, 2001*). Problem-based learning (PBL) and case-based learning (CBL) have emerged as powerful tools in reforming traditional teaching methods. PBL in medical education uses the patient's problem as a stimulator for students to learn problem-solving skills while CBL is a group discussion-styled teaching approach based on analysis of authentic clinical cases (*Tayem, 2013*; *Jackson, 2003*; *Donner & Bickley, 1993*; *Finucane, Johnson & Prideaux, 1998*). PBL and CBL engage students in their own learning, focus on concrete scenarios like problems or cases, and emphasize the development of thinking skills (*Hofsten, Gustafsson & Haggstron, 2010*; *Chan, Hsu & Hong, 2008*; *Hakkarainen, Saarelainen & Ruokamo, 2007*). They comply with the key elements of adult learning theory (*Nadershahi et al., 2013*), making them the promising instructional methods to teach MSFA.

However, CBL or PBL present some limitations in teaching MSFA if applied alone. First of all, CBL may not provide an organized view of knowledge as it situates knowledge in real-world contexts in a piecemeal way. The students, who usually do not have pre-established knowledge, may find it difficult to learn a new subject using CBL method alone (*Williams, 2005*). On the other hand, as it requires students to learn background knowledge by solving problems during the class session, PBL is effective for students who don't have pre-established knowledge provided the problems are properly framed (*Williams, 2005*). However, the teacher who poses a problem without cases or context may find it difficult to frame the problems and engage students' interests. Cases can help contextualize the problems and framed the knowledge in a logical and organized way (*Allchin, 2013*). PBL is primarily student-driven whereas CBL is collaborative (*Williams, 2005*), which means the teachers can be more intimately and directly involved by CBL, making it easier for them to frame and contextualize the problems. Therefore, PBL and CBL are mutually complementary. PBL can amplify the basic virtues of CBL, while CBL can facilitate framing

and contextualizing the problems. This study is to combine the two methods and evaluate the efficacy of the combined PBL-CBL method in teaching MSFA.

# MATERIALS & METHODS

## Students

This study was conducted according to the guidelines set forth by the Declaration of Helsinki and approved by the Ethics Committee of the Affiliated Stomatological Hospital of Chongqing Medical University (No. KQJ201816). Written informed consent was obtained from all students. Ninety-two clinicians who received training between 2015 and 2017 at the Department of Implant Dentistry, the Affiliated Stomatological Hospital of Chongqing Medical University, were included in this study. All the students were junior doctors, aged between 25 to 30 years, and were granted with a full-time undergraduate degree from dental colleges in China. No students had any experience or training in MSFA. The students were randomly allocated into an experimental group and a control group. In the experimental group, the students (25 males and 21 females) were trained using the PBL-CBL method, while the students in the control group (24 males and 22 females) were trained using the traditional teacher-centered method. Both groups were trained for a period of 3 months.

## Teaching methods

All students attended class sessions on five topics for MSFA, "1. Anatomy of Maxillary Sinus", "2. Pre-surgical Assessments and Treatment Plan", "3. Surgical Principles and Procedures", "4. Points for Attention during MSFA", and "5. Management of Complications". The curriculum of MSFA was completed in eighteen sessions, with each session lasting for forty minutes. In the control group, the students sequentially attended the sessions in the form of teacher-centered lectures, and the role of the teacher was to dispense final form knowledge. There was no scheduled discussion time during or beyond the class session. In the experimental group, the students attended no formal lectures. Instead the students were divided into small groups of 3 or 4 members. Discussions about the topic were held in each of the 18 sessions. The role of the teacher shifted from conventional authority to a case narrator and an expert guide for discussion. The total duration and number of sessions were the same in the two groups.

The teaching method in the experimental group is described as follows to show how CBL and PBL were combined. The parenthetical abbreviations at the end of the sentences, namely (PBL) or (CBL), indicates that the activity or method described in the sentence is drawn from PBL or CBL.

### Assignment of pre-class work and introduction of typical cases and problems

Before the class session started, the teacher asked the students to do pre-class work related to the topic of the course, such as searching and reading information in papers, books or on authorized websites (PBL). The teacher prepared one or more typical clinical cases in advance to engage the students' interests on the topic (CBL). During the class session, the

teacher presented the cases to provide the students with detailed information about the patient's chief complaint, history of present illness, medical history, intraoral examination, cone-beam computed tomography scan and the research plaster model (CBL). The teacher would interrupt the case by raising problems related to the topic of the course (PBL). The students were asked to use existing clinical data and discuss in small groups. Then each group made comprehensive analysis, proposed effective treatment plans, analyzed possible risks and identified ways to avert such risks, and explained their reasons (PBL).

### Group reports and Q&A session

At the last session of each of the five topic, the teacher commented on the outcome of discussions reported by each group (CBL) and relevant questions were raised by both the teacher and the students (PBL). The teacher guided the discussion on some questions while leaving the others for the students to think about (PBL). These were open-ended questions that would arouse the students' interest in learning and encourage them to further explore the issues (PBL). Questions about the next topic were raised and framed in the context of cases by the teacher to cue the need for background knowledge (CBL and PBL). The students would then begin to search for and read materials related to these questions and make preparations for discussion in the sessions of the next topic (PBL). Meanwhile, the teacher would offer guidance to the students on how to retrieve information online or from the library (PBL).

### Summary of MSFA and development of a treatment plan for a complex case

At the final session, the teacher presented a complex case and raised questions on the five topics. The students were asked to discuss the case in groups and make a treatment plan. The group leader then summarized their discussion and presented a summary on behalf of the group members. The teacher analyzed and summarized the key points and determined the final treatment plan together with all the students.

## Evaluation methodology

The outcomes of different teaching methods were evaluated in the following three ways. Two teachers from the department of implantology graded the exams. The graders were blinded to the name of the students and the group they belonged to.

### Anonymous questionnaire

After the training, anonymous questionnaires made by the researchers compromised of nine questions were filled out by both the students from the experimental group and the control group. The detailed information of the questionnaire was revealed in Table 1.

### Theory test

At the end of training, students took the final exam which included four questions, namely indications for sinus augmentation, preoperative assessments of MSFA, procedures for MSFA, and management of maxillary sinus membrane perforation. The total score were 100 points, 25 points for each question.

**Table 1  Opinions on the PBL-CBL Method (PBL-CBL) and the teacher-centered teaching method (control).**

| Items surveyed | Rate of satisfaction | | P value |
|---|---|---|---|
| | PBL-CBL | control | |
| 1. I like this approach | 91.3% | 76.1% | 0.0482 |
| 2. This approach is efficient | 89.1% | 73.9% | 0.0601 |
| 3. This approach decreases extracurricular work | 65.2% | 87.0% | 0.0145 |
| 4. This approach makes learning more targeted and more interesting | 95.7% | 60.9% | <0.001 |
| 5. This approach enhances my ability to analyze and solve problems | 93.5% | 39.1% | <0.001 |
| 6. This approach helps me master theoretical knowledge | 87.0% | 82.6% | 0.5616 |
| 7. This approach helps me improve clinical skills | 95.7% | 52.2% | <0.001 |
| 8. This approach facilitates clinician-patient communication | 87.0% | 73.9% | 0.1148 |
| 9. This model emphasizes more on teamwork | 93.5% | 32.6% | <0.001 |

*Case analysis*

After the theory test, the teacher presented a new case which was different from the cases discussed earlier in the class. The teacher provided the students with detailed information about the patient and the students were required to answer a series of questions about the key points that had been taught or discussed in the class sessions in a written form. Finally, the papers were graded. The test paper and the scale of marks were attached as Supplemental Information 2.

## Statistical analysis

Pearson's chi-squared test was used to analyze the difference in gender and the students' opinions about the teaching methods between the experimental group and the control group. The scores of the theory test and the case analysis were expressed as mean ±standard deviation (SD). One-way ANOVA was applied to analyze the difference in scores between the experimental group and the control group. All tests were two-sided, and $p < 0.05$ was considered significant. Statistical analyses were performed using the statistical package SPSS (version 20.0, IBM, Armonk, NY, USA).

## RESULTS

A total of ninety-two students (49 men and 43 women), aged between 25 and 33 years (mean:28.6 years), were included in this study. No students were lost to follow-up. There was no significant difference between the control group and experimental group with regard to gender ($p = 0.883$). All students followed the schedule and attended the lectures or discussion on time. Table 1 shows the survey results of the questionnaire. The rate of satisfaction with the PBL-CPL method is higher than that with the control method in all the items except for "This approach decreases extracurricular workload".

Table 2 shows the scores of the theory test and the case analysis. Compared with the teacher-centered approach, the PBL-CBL method resulted in higher scores in both the

**Table 2 Comparison of average scores of the two groups ($n = 46$).** The experimental group was exposed to the PBL-CBL method; control group in the traditional teacher-centered curriculum.

| Group | Gender | Theory test | Case analysis | Total score |
|---|---|---|---|---|
| Experimental group ($n = 46$) | Male: $n = 25$ | $80.69 \pm 3.25$ | $76.30 \pm 3.01$ | $78.50 \pm 3.21$ |
| | Female: $n = 21$ | | | |
| Control Group($n = 46$) | Male: $n = 24$ | $76.34 \pm 3.46$ | $72.19 \pm 2.82$ | $74.27 \pm 3.07$ |
| | Female: $n = 22$ | | | |
| $F$ Value | | 38.432 | 45.443 | 40.304 |
| $P$ | | <0.01 | <0.01 | <0.01 |

theory test and the case analysis. The score differences between the two studied groups were statistically significant for both the theory test ($P < 0.01$) and case analysis ($P < 0.01$). The students in the experimental group presented a generally better understanding of MSFA based on the theory test and case analysis.

## DISCUSSION

PBL and CBL have been described as promising tools for medical and dental education and have been used in varied fields of dental education (*Donoff, 2006*; *Major & Palmer, 2001*; *Wang et al., 2008*; *Koh et al., 2008*; *Thistlethwaite et al., 2012*; *Tomaz et al., 2015*). To maximize the effect of PBL and CBL, our study applied a teaching method integrating PBL and CBL in teaching MSFA, which achieved higher efficacy than the traditional teacher-centered method. According to the students' feedback, more than 90% of the students believed that the combined method made learning more targeted, enhanced their problem-solving ability, improved their clinical skills and raised their teamwork awareness. These results are consistent with the results of other studies in which the combined method was applied in leadership training or biochemistry experiment teaching (*Dong & Zeng, 2017*; *Ginzburg et al., 2018*). In general, the PBL-CBL method was shown to be more effective than the teacher-centered method in teaching MSFA. Combined PBL-CBL may be a useful model for teaching complex oral surgery in dentistry.

Previous studies pointed out that CBL was not effective in conveying existing knowledge system, which was typically conveyed in didactic teacher-centered approached (*Allchin, 2013*; *Jamkar, Yemul & Singh, 2006*). In this study, we interrupted cases with a series of well-contextualized questions or problems. Then the students were asked to use reference book, library and online resources to solve the problems. In this way we contextualized the knowledge in authentic cases and embodied the rationale for learning by posing problems. Previous studies reported that PBL was able to cover approximately 80 percent of what could be accomplished in a didactic approach in the same period (*Albanese, 1993*; *Berkson, 1993*). The result of the theory test in our study showed that students in the PBL-CBL group had formed a comprehensive and organized understanding of MSFA. The PBL-CBL method was even advantageous over the didactic approach in conveying existing knowledge system, suggesting that problems well-framed in cases could cover standard curricular content.

In addition to basic knowledge, the result of case analysis further showed that students in the PBL-CBL group were more likely to use the acquired knowledge spontaneously to solve new problems than those who acquired the same information through lectures. The rate of satisfaction with the PBL-CPL method was higher than that with the control except for the item "This approach decreases extracurricular work". Although this was only a subjective feeling of the students, it did show that the PBL-CBL approach was positively accepted among students. Students who thought the PBL-CBL approach did not decrease extracurricular workload may have spent more time searching for information. Therefore, in order to take full advantage of the PBL-CBL methodology, the faculty members should be trained more vigorously to lead discussion groups and provide assistance to develop the students' capacity in searching for and generalizing information.

There were still some imitations in study design and methodology in this preliminary study. We assumed that PBL and CBL were mutually complementary and could achieve the best effect when combined. However, no control groups using PBL or CBL alone were included in this study, and we were not able to determine whether the hybrid method was superior to PBL or CBL alone. To gain feedback about the hybrid method, a Yes/No scale was used in the questionnaire, which only resulted in rough calculations. A Likert Scale would be more appropriate and accurate to scale responses and detect difference in survey research. In addition, one study was inadequate to prove the efficacy of the PBL-CBL method. Further randomized controlled trial was needed to confirm the effect of the PBL-CBL method.

## CONCLUSION

The students learning MSFA with the PBL-CBL method exhibited better acquisition of academic knowledge and higher competence in case analysis compared with those learning MSFA with the traditional teacher-centered method. This research suggested that the PBL-CBL method be a promising new mode for teaching complex surgical techniques in implant dentistry and other dental fields.

### Funding

This work was supported by the Education Reform Research Project of Affiliated Stomatological Hospital of Chongqing Medical University (No. KQJ201816), the Program for Innovation Team Building at Institutions of Higher Education in Chongqing in 2016 (No.CXTDG201602006) and the Scientific Research Project of Chongqing Municipal Commission of Health (No.2018MSXM124). The funders had no role in study design, data collection and analysis, decision to publish, or preparation of the manuscript.

### Competing Interests

The authors declare there are no competing interests.

## Author Contributions

- Yunfei Liu conceived and designed the experiments, performed the experiments, analyzed the data, prepared figures and/or tables, authored or reviewed drafts of the paper, and approved the final draft.
- Yamei Xu performed the experiments, analyzed the data, prepared figures and/or tables, and approved the final draft.
- Yueheng Li and Qingqing Wu conceived and designed the experiments, authored or reviewed drafts of the paper, and approved the final draft.

## Human Ethics

The following information was supplied relating to ethical approvals (i.e., approving body and any reference numbers):

The Ethics Committee of the Stomatological Hospital of Chongqing Medical University granted ethical approval to carry out the study within its facilities (Ethical Application Ref: KQJ201816).

## Ethics

The following information was supplied relating to ethical approvals (i.e., approving body and any reference numbers):

Stomatological Hospital of Chongqing Medical University granted Ethical approval to carry out the study within its facilities (Ethical Application Ref: KQJ201816).

## Data Availability

The raw data are available in the Supplemental Files.

## Supplemental Information

Supplemental information for this article can be found online at http://dx.doi.org/10.7717/peerj.8353#supplemental-information.

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
