# Peer review of "Application of problem-based learning and case-based learning integrated method in the teaching of maxillary sinus floor augmentation in implant dentistry"

_PeerJ, doi:10.7717/peerj.8353_

## Round 0.1 · original submission · Major Revisions

Please pay careful attention to the comments of the reviewers and respond to all of the criticisms. In particular, all three of the reviewers suggested that there are problems with the distinction between PBL and CBL in the manuscript, and Reviewer 1 believes that your definition of PBL is not entirely correct.

·

Basic reporting

overall, the writing is clear and well-written, with professional English used throughout. The descriptions of PBL and CBL (and distinctions between the two) are ambiguous and not entirely correct or consistent with some of the literature.

I provide the authors with the following comments and suggestions:
– I believe that the description of PBL is mischaracterized, to some extent. The claim in line 49 on page 1 - “this teaching method is based on questions” is not precisely true. The socratic method is based on asking and answering questions. Problem based learning is a student-centered approach to problem solving. Asking questions may or may not be part of the methodology. Barrows, 1996 provides the definition of problem-based learning and outlines the main characteristics of the methodology. Barrow points to authentic problems as the basis for PBL, not necessarily questions. PBL is also, in its strictest form, self-directed and student driven, with students working together in small groups. In your introduction the suggestion that the teacher is leading students to examine and answer questions in a step by step manner is actually contrary to the characteristics of PBL outlined by Barrows. There are numerous modified versions of PBL out there (published) but if you are using a definition of PBL that is different from that of Barrow you should provide the citation and explicitly state how it is different from Barrow’s. You should also explain what you mean by “the traditional approach” (line 55) as this could be interpreted many different ways.

You should also make a stronger comparison between PBL and CBL in the introduction. These are two different methodologies. Currently, the description you include of CBL is very much like the description of PBL. It would be helpful to indicate how they are different and mention some contexts in which one is used over the other. Nadershani et al. 2013 provides a good example of how to do this (in the context of dental education) and includes many sources that you might find relevant. Nadershahi, N. A., Bender, D. J., Beck, L., Lyon, C., & Blaseio, A. (2013). An Overview of Case-Based and Problem-Based Learning Methodologies for Dental Education. Journal of Dental Education, 77(10), 1300–1305.

Some sentences could be rephrased for clarity: for example, the paragraph starting at line 137 needs to be reworded. The second sentence beginning with “Besides, the difference of theory test…..” is unclear and I’m not sure if you are trying to say that the results for the case analysis were also significantly different?

Experimental design

I appreciate the sample size in the study and that gender of participants was included. Gender should also be included in Table 1. If there are data on level of experience of participants that should also be included. Currently there is no assessment of baseline knowledge for either control or experimental group. The authors also do not explain why some of the choices in study design were made. For example, I am unsure why you chose to combine PBL and CBL into a new methodology rather than selecting to use one or the other. I don’t think there is anything wrong with using a modified version or a combined version but you need to provide justification for it, describe how the new method is different from both original methods, and include a discussion of why the new method you use is appropriate.

The authors also could provide more detail about the rationale for selecting maxillary sinus floor augmentation for their case: what is it about maxillary sinus floor augmentation that makes learning these skills particularly challenging? Why is this more challenging than learning other skills in implant dentistry? What evidence exists to support your claim that this is more challenging than learning other skills?

Some other comments about the study design and reporting of the study design:
Line 99 – teaching methods. You should include a description of the control group activities. If they didn’t participate in discussions did they simply attend lectures? Were these lectures in person or online? Did the experimental group also attend lectures, in addition to the discussion sessions or did they attend discussion groups only?

Sections 2.2.1, 2.2.2, 2.2.3 - it is not clear whether ALL students engaged in these activities or if these were just activities in which the experimental group participated. You should clarify the differences in learning experience between the control and experimental group. A chart or table would be a nice way to demonstrate this.

In your description of these activities, it also seems that the teacher is doing a larger percentage of the instruction and directing and that there is very little self-directed learning on the part of the students. If you are claiming that this methodology is a blend of PBL and CBL you should highlight the components of these activities that reflect the PBL and CBL experiences.

Line 129 - I’m not sure what you mean by “the final exam, which was made of the theory test and case analysis”. Can you describe the exam? How many questions, what were the formats for the questions, what content was covered, is this a validated exam, was it piloted with any other group or is it an exam that has been given previously to other cohorts?

Line 134 - more information should be provided about the questionnaire given at the end of the training. How many questions, what were the formats for the questions, what content was covered, is this a validated questionnaire, was it piloted with any other group? It would be helpful to include a copy of the questionnaire as supplemental material with the manuscript

Validity of the findings

Line 142 – you need to include a summary of the results from the questionnaire rather than just directing the reader to the table. What were the major results? Did you perform any sort of analysis on these results? If you did not analyze them quantitatively or qualitatively, then you should not report or mention them in this manuscript.

Line 146 – I think that the section 3.3 is referring to data obtained from the questionnaire but this is unclear. You should specify from where these conclusions are being drawn. It also seems that you are reading quite a bit into the results of the survey. Was any sort of analysis conducted? It is possible to conduct quite robust analyses of qualitative data.

Discussion – line 151 – is there evidence to support your claim that implant dentistry is growing rapidly? You should include statistics to support this, as well as other statements you make about the presence of absence of implant clinics.

In line 189 you claim that students were more likely to actively go about clinical work with questions in mind and were motivated to solve questions etc.. but this was never specifically tested in your study and there are no data to support these claims. Either make it clear that this is anecdotal and that it has not been tested so you do not know whether this is truly the case, or provide evidence to support these claims.

Additional comments

in addition to comments above, you need to make clear what you mean by “traditional method” hroughout the manuscript. Also - the discussion section largely reiterates many points from the introduction. The entire section in the discussion on PBL and CBL (lines 160-180) could be moved and combined with the introduction to make that more robust. You discussion section should discuss the significance of your results, the limitations of your study (this is currently not included), and what your results mean in the context of other existing literature on the topic.

Reviewer 2 ·

Basic reporting

a. Clear and unambiguous, professional English
i. Line 26: Ninety-two clinicians who were receiving training between 2015 and 2017…should be Ninety-two clinicians who received training…
ii. Line 31: After three months of training, teaching performance of both groups was evaluated through a theory test, case analysis and a…what do you mean by teaching performance? This implies that the participants
iii. Line 45: instead of “by now”, use “currently”
iv. Line 70: typo: toadvances should be to advances
v. Line 72: “Implant dentistry is an emerging independent branch of dental science and it involves knowledge of many other dental fields such as” may be better stated as “implant dentistry is an emerging independent branch of dental science which involves
vi. Line 75: what do you mean by social recognition? Perhaps social is the wrong word here, perhaps “greater recognition in the field of dentistry”?
vii. Line 76: training in implant dentistry
viii. Line 77: why is it of great urgency to find out how to enhance theoretical knowledge and basic skills of clinicians?
ix. Line 91: should be: ninety-two clinicians who received training…were divided randomly and equally…
x. Line 103: discussions were held every other week and lasted 3 class sessions every time. Do you mean 3 class sessions every other week? How long was each class session, or how long on average did each class session occur?
xi. Line 107: This clinical case was a case on an adaptation syndrome…
xii. Line 110: spell out Computed Tomography (CT)
xiii. Line 116: leaving some, not leavingsome
xiv. Line 118: explore the, not explorethe
xv. Line 118: The students would then begin to search…
xvi. Line 126: largely encouraging so as to
xvii. Line 127: students develop
xviii. Line 127: thinking. Not thinking,
xix. Line 129: final exam, which…not final exam,which (space after comma)
xx. Line 138: control group, respectively. Not control group , respectively
xxi. Line 156: clinical practice. Not clinicalpractice
xxii. Line 162: over-reliance on. Not over-relianceon
xxiii. Line 169: develop an
xxiv. Line 186: literature and to propose. Not literatureand topropose
b. Introduction and Background
i. Introduction
1. More detail is needed on the benefits and theories of PBL. For example…” many schools desired to follow McMaster University, where the model for student-centered, problem-based, small-group learning took shape. The faculty and administration at McMaster University had found that the PBL approach to learning encourages students to direct their own learning and focus on what they need to know to solve real-life problems, to interact with others and work as a team to solve problems, to describe knowledge and ideas in a manner that can be understood by other team members and to use an existing knowledge base to solve new problems in a creative manner (Schmidt, H., Hermans, H., Venekamp, R. The Development of Diagnostic Competence: Comparison of a Problem-based, an Integrated, and a Conventional Medical Curriculum. Academic Medicine 71(6):658-664, 1996.)
2. Line 52: What is the step by step manner to examine and answer these questions? How does this aid in learning the subject content?
3. Line 55: there are many other important comparisons to the benefits and advantages of PBL over a traditional didactic teaching approach. For example…” A study was performed to elucidate the goals of PBL programs in different institutions. Goals common to all programs include the desire to: stimulate development of communication skills; emphasize concepts and interdisciplinary biomedical principles required for the understanding and management of health problems; provide for learning in context for the development and practice of reasoning skills that will be used throughout life; develop skills for management of information; foster active, self-directed learning, and cultivate the personal qualities required for a career in the health professions including a sensitivity to the needs of others and responsiveness to ethical and moral issues. (Herron, M., Whitney, M., Weeks, B. Preclinical Curricular Alternatives: Goals and Methods of Problem-Based Learning. Journal of Veterinary Medical Education 20(1):14-18, 1993.”…and “The comparison of an innovative pedagogy in an institution and its former conventional program or a separate conventional didactic program seems like a natural one, assessing the merits of a new curriculum against those of the old format. There is evidence that PBL students retain knowledge much longer than students taught conventionally. In studies requiring integration of basic and clinical knowledge, problem-based students tend to do better in providing causal explanations of pathophysiologic processes underlying disease (Camp. G. Problem-Based Learning: A Paradigm Shift or a Passing Fad? Medical Education Online 1:2, 1996.)”
4. Line 58: Both PBL and CBL are described separately. What are the similarities and what are the differences? PBL is case based, but CBL is not necessarily problem based learning.
5. Line 75: What do you mean by social recognition? Why does implant dentistry need to gain greater social recognition? Do you mean that implant dentistry is gaining greater recognition as a legitimate dental specialty?
6. What is the traditional teaching approach at this program? A brief but detailed explanation is necessary to compare to PBL-CBL

c. Figures: Table 1: in the description of the table, briefly describe each group ie, the two groups (experimental group exposed to the PBL-CBL method; control group in the traditional didactic curriculum)
i. In the top row, the Score of Theory test should be relabeled as Average Score of Theory Test. Same for Score of case analysis

Experimental design

2. Experimental Design:
a. Line 111: What comprises a comprehensive analysis? What does this mean?
b. Line 123: how were group leaders determined? What were their roles?
c. Line 123: What are some examples of how the group leaders summarized their discussions and presented summaries?
d. Line 127: how does this relate to learning?
e. Line 127: what was the other method of teaching and learning? Compare and contrast
f. The experimental design is basic, but appropriate for comparing the experimental group and control group.
g. Line 129: more explanation of the exam. What do you mean by “it was made of the theory and case analysis”?
h. Line 143: summary of questionnaire, examples of questions and responses
i. Line 147: what roles were shifted of the teacher in the classroom as a result of participating in this method? Where is the questionnaire for the teachers?

Validity of the findings

3. Validity of the Findings:
a. Line 153: reference for dentists attending specialized training, or is this the authors’ opinion?
b. Line 156: reference for this statement?
c. Line 188: example of the developing and use of a clinical case? Examples of a main logical thread, key theoretical knowledge, examples of reasonable treatment plans and teacher summaries?
d. Line 187: where did the teacher summarize the plans from different groups? In a lecture format? And why did the teacher do this? What were the benefits of this?
e. Line 189: students were more likely to actively go about their clinical work with questions in mind: what is the evidence of this? How was this evaluated?
f. Line 190: they were motivated to solve these questions in their clinical work instead of conducting clinical operations passively; they put theories into practice in a more flexible way…where is the evidence of this? How was this evaluated? Was it the teachers’ observations? Students’ self-perceptions? How could one tell the difference in clinical work between students that had gone through PBL-CBL and those that had gone through traditional teaching and learning methods?
g. Conclusion: students exhibited increased level of theoretical knowledge and higher ability of case analysis: the increased level of theoretical knowledge was exhibited by higher test scores, but how can you conclude that these students had a higher ability for case analysis? There were no research questions on this topic, and no data to support it.
h. In Table 2 the evaluation of PBL-CBL method by the experimental group, several of the items surveyed seem a bit ambiguous in exactly what was being asked and answered. For example, the item “this approach enhances my ability to analyze and solve problems”…what methods were employed such that the students actually knows what this means? Were the guided in ways to analyze and solve problems? By which criteria?
i. Also in the table of items surveyed, students voted yes or no on “this approach helps me master theoretical knowledge”…the students probably should have been asked about ability to master theoretical knowledge compared to a traditional didactic curriculum, such as efficiency, ability to recall in the long term, etc.
j. And, a Likert Scale would have been much better to use, rather than Yes/No
k. In the discussion, limitations of the study need to be discussed. Such as that one course in PBL-CBL may not be enough to fully have students realize the benefits of PBL. Also, the lack of a Likert Scale.

l. Conclusion: what are the implications for this research, future directions, benefits, other applications, etc. This conclusion is too brief. How can you conclude that the students have a higher ability of case analysis? How was this measured? There is no evidence of this in your data. The only research question that was answered was the comparison of theoretical knowledge between the two groups via examination.

Additional comments

4. Overall, while I also have the opinion that students learn theoretical knowledge better and have better long-term recall, have developed critical thinking and problem solving skills, do better in group learning situations and develop teamwork and communication skills, learn how to analyze and process cases better, in this paper there doesn’t seem to be the evidence and data to support anything other than that the experimental group performed better on one exam in theoretical knowledge than the control group. While that in itself has much validity, any conclusions drawn from that are speculative. This paper does however discuss applying PBL-CBL to a specific part of a curriculum in which it has not been done before, so there are benefits in discussing that in this paper.

·

Basic reporting

There are some minor improvements that can be made to the document to improve the language for the reader's understanding. Specifically, the sentence on lines 138 through 140 that begins with the word "besides' is ambiguous in meaning. Can you please clarify this sentence, and verify that "Besides" is the appropriate word usage?

Experimental design

In the Introduction, a clearer distinction between the CBL and PBL approaches to learning is need to identify how the two approaches differ and how they are similar. In addition, how are these two approaches integrated? Is it that both are utilized in their entirety, or is there some hybrid combination of the approaches that only uses certain aspects of each? Please clarify.

In the section on Teaching Methods, can you please provide data concerning the amount of time that the control group met in oder to add verification that the differences in learning outcome are not just due to the amount of time that was spent by each group interacting with the material?

In the Materials and Methods, can you please provide more information about the content and structure of the Theory test and the Case analysis so give the reader a clearer view of how your data on performance was collected?

Some additional discussion of the results from the questionnaire on line 142 is needed. Was the control group given a questionnaire as well to gauge their response to traditional teaching methods?

Validity of the findings

no comment

---

## Round 0.2 · Major Revisions

Both Reviewer 1 and Reviewer 3 believe that your manuscript still requires important changes and that problems from your original submission remain. In particular, the distinctions you make between PBL and CBL is still not clear to either of those reviewers.

In addition, questions about the statistical analysis have not been addressed sufficiently. You state that t-tests and ANOVA's were performed, but only two tests are reported. F-distribution values are given for those results, [difference of theory test (Degrees of freedom: 91, F=38.432, p=0.001<0.05) and case analysis (Degrees of freedom: 91, F=45.443, p=0.001<0.05)], but the Degrees of Freedom (91 or n-1) do not seem to be correct for an ANOVA comparing the means of the two groups. Were these Student's t-tests or ANOVA's? Please provide a much more detailed description of the statistical tests that were performed and the results reported.

In your re-submission, please be careful to respond to my comments above as well as all of the comments of Reviewer 1 and Reviewer 3.

·

Basic reporting

There are several minor edits to English grammar and phrasing needed throughout; this would help with overall flow and clarity (for example, check the tense used throughout the paper - within sections or paragraphs this should be consistent). Some citations are not in English - be sure that all citations are translated.
Line 81 in the introduction "problem-based students" should probably be changed to "students in problem-based learning classrooms/activities"... something similar.

Line 90 - the description of CBL is a little confusing to me. CBL does not necessarily require that students simulate doctors' sessions with patients.

The last paragraph of the introduction (on Implant Dentistry) seems out of place or it seems that there is a clunky transition to this paragraph. Consider adding subheadings or sections to the introduction to help with the flow of the narrative.
Line 118 - it would be helpful to include a sentence explaining why maxillary sinus augmentation is particularly difficult to teach (compared to other procedures) - does it require greater background knowledge than other procedures? is it just a more complicated procedure? are there novel technical skills that must be mastered?

Experimental design

Line 129 - what do you mean by "there were no differences in learning experiences"? It's highly unlikely that all participants have identical learning experiences unless they all attended the same institutions throughout their entire educational career- surely there are some differences. it might be better to state how they are similar - for example, all participants hold a DDS or DMD degree or have participated in a residency program, etc.

Line 138 - What is "matters that warrant special attention"? is this the name of a course?
I have a much better idea of the differences between the control and experimental groups but I'm still confused about some aspects of the study design:
lines 142-144 - you state that the amount of time spent by each group in training is equivalent. do you mean 40 min/session or do you mean the number of sessions (or both). how many total hours of instruction time/training team did each group get? it is not clear from the current text whether this is truly equivalent or not.

it is still not clear to me which aspects of the activity draw from PBL and which draw from CBL (in fact, it seems to most closely align with CBL) - the authors should call these out specifically since they claim that the training method is an integration of both methodologies. Instructors at other institutions who may wish to replicate such a training process should be able to understand exactly how this activity integrates PBL and CBL.

the results section indicates that the final exams consisted of short answer questions - how were these graded in order to ensure consistency in grading and how were points determined? is there a rubric that was used?

for the case analysis - was this a novel case or one that the students saw previously? again - how were students graded? is there a rubric?

line 196-198 - I'm not sure what you mean by "the difference of theory test (Degrees of freedom: 91, F=38.432, p=0.001<0.05) and case analysis (Degrees of freedom: 91, F=45.443, p=0.001<0.05) was statistically significant."

Lines 217-224 need to be rewritten for better flow and clarity. I would include a subheading for "limitations of the study"

Validity of the findings

line 189-192 - was a t-test or ANOVA used? why were both tests used and which variables were subjected to each statistical test? a more robust description of the statistical analysis is required.

again - it is never explained in the manuscript how this approach integrates PBL and CBL. the authors should specifically call out which characteristics of PBL are reflected in their approach and which characteristics of CBL are reflected in their approach. this could be done easily with a chart or table. This is essential as one of their main points in the discussion is that this is "this study was the first to apply a teaching method integrating both PBL and CBL in a specific curriculum..."

Conclusions can also include a statement about how this might be relevant for other educators or implemented at other institutions.

Reviewer 2 ·

Basic reporting

no comment

Experimental design

no comment

Validity of the findings

no comment

Additional comments

i have reviewed the changes made to this manuscript based on my suggestions from my first review. I am satisfied with the improvements made by the authors.

·

Basic reporting

1) Discussion Notes (Section 4):
a. Lines 220-224 with the sentence beginning “Besides…” does not read smoothly and should be reworked to be clearer in their intent to convey information
b. Line 227 beginning with “More…” Could be better stated as “Further research is needed to confirm this”

2) Materials and Methods Notes (Section 2)
a. Line 129 “There were no differences in learning experience…” Do you mean that there was no difference in the educational background between the control and the experimental groups?
b. Line 139 In place of “Totally”, I believe you mean “In total”
c. Line 139 Do you mean to say that every session lasted 40 minutes?
d. Line 140 “Beyond that…” Does “that” refer to the class sessions?
e. Line 143 Do you mean to say that every session lasted 40 minutes?
f. Line 144 The sentence “In addition…” could just be changed to “The experimental group did not attend any lectures”

3) Introduction Notes (Section 1)
a. Line 81 The citation is not in English
b. Line 87 The sentence beginning “On the other hand…” is unclear in its meaning and should be reworded
c. Line 117 Why is teaching maxillary sinus floor augmentation particularly challenging? Please include an explanation in the text.

Experimental design

2) Materials and Methods Notes (Section 2)Lines
a. 146-168 In your description of the experimental group, it would be helpful to explicitly state which portions of their class design is taken from PBL and which is derived from CBL
b. Lines 147-154 and 176-181 are identical, and it appears that the first describes the discussion design and that the second describes the clinical exam that the students were given
i. Was the clinical exam written or oral?
ii. Are the two exactly the same? Is the case the same that had already been discussed in the class, or was the exam a novel clinical scenario?
iii. On line 175, it would be helpful to add “exam” to the subject heading for clarification

Validity of the findings

The discussion still lacks an analysis of the results within the context of existing literature on the topic.

---

## Round 0.3 · Major Revisions

Although the reviewers appreciate that you have made some revisions, both of them believe that there are still significant flaws in your manuscript. If you decide to submit another revision, please be sure to address all of the issues raised by the two reviewers.

·

Basic reporting

Please use consistent tense throughout the manuscript – currently needs more editing for grammar and English language.

Be consistent with use of either PBL-CBL or CBL-PBL throughout the manuscript. Both are used in various places – pick one and stick with it.

The discussion of MSFA and the particular challenges of teaching this topic is much improved.

The discussion of PBL and CBL, their similarities and differences is much improved.

The last paragraph of the Introduction (right before Materials and Methods) needs to be rewritten for better English grammar and wording.

Throughout the manuscript you mention that you are evaluating the “efficiency” of this teaching method. Do you mean “efficacy” instead of “efficiency”? if you mean efficiency is an outcome, you should have some measurement of time on task or time for completion.

ABSTRACT: There is not a real hypothesis in the abstract. This may be okay as the research may be exploratory, in nature, however surely the authors went into the study with a hypothesis about the method – it would be good to include it. Also – in the results section of the abstract you don’t indicate what you were testing with the theory test and case analysis. Was this all satisfaction with the method? In the conclusion for the abstract, as well – the application of the methods shows positive results for what? Learner satisfaction? For improved learning? Be clear about what the findings mean for the reader. The abstract needs to give a reader a condensed version of the study so that they can quickly determine what you did and why you did it. Please revise the abstract to include the essential details of the study.

I see that in the Materials and methods section you attempt to indicate which elements are drawn from PBL and CBL by using (CBL) and (PBL) but you do not indicate in the manuscript that this what you are doing. You could provide general statements about the ways in which your method draws from each of these. If you want to identify the specific parts of each method and label them as (CBL) or (PBL) you need to indicate this in some sort of introductory sentence or two before you do it, otherwise readers will not know what the parenthetical abbreviations are supposed to indicate. My personal preference would be to simply indicate all this in a chart or diagram but if you do it in text, be sure you clearly explain what you are doing to the reader.

Description of statistical analysis needs to be rewritten for English language and grammar.

Experimental design

Did the traditional group not complete a questionnaire soliciting their opinions on that approach? If not – why not? I’m not sure the questionnaire results are meaningful if you only have satisfaction scores from the experimental group. It’s possible the traditional group would also have revealed a high satisfaction rate, too. You also make the claim in your discussion that PBL-CBL method gained popularity among the students but if you only allowed the experimental group to express their thoughts about teaching and learning methods you haven’t actually captured the perceptions of the rest of the class.

This sentence before the Discussion section is inaccurate: The participants in the experimental group present a generally better understanding of the background knowledge and the key points when performing MSFA. Unless the learners were observed performing MSFA and there are data on observed performance, you don’t know that this is true. Your results may show that the experimental group performed better on exam questions about MSFA but you don’t have data on how this translates to actually performing the procedure.

also – in the discussion, the data may support your hypothesis that this method was more effective but one study doesn’t prove that. Be careful with your wording.

Also- I still don’t understand why teaching efficiency is your outcome. Are you suggesting that the method lets you reduce the number of teaching hours for the topic so that it is more efficient? Or that learners learn to perform the procedure more quickly so it is more efficient? “efficient” implies that there is a more rapid rate of productivity. If you mean “effective” of “efficacious” than please change this throughout the manuscript.

Is there a rubric for grading the case analysis? If so – please include it. It would also be helpful to see some of the questions that are included in the case analysis.

Validity of the findings

This sentence in the methods: “These comments were largely encouraging so as to help the students develop a scientific way of thinking.” This is an assumption unless you cite literature to support that providing encouraging words helps promote scientific thought. Please provide citations.

The discussion section still has many paragraphs that would be more appropriate in the introduction section. The discussion should simply contextualize your results in the existing literature. What do your findings contribute to this body of work? How do your results support or refute what is already known? What is significant about what you found?

I appreciate that a section on limitations is added but you should include some text describing why the limitations you mention are limitations. Currently, it feels like it was tagged on at the end without consideration for how readers might be able to take these points as suggestions for future work.

Conclusion – again, as in the abstract, the application showed positive results – for what? For student satisfaction? For retention of knowledge? For application of knowledge? Your hypothesis is not clear.

Additional comments

All comments are indicated in the areas, above.

·

Basic reporting

The authors have done well in addressing many of the issues that were pointed out in the previous reviews. The justification for focusing on Maxillary Sinus Floor Augmentation is included. Also, the explanation of CBL versus PBL, and the way in which they were combined for this study, provides much more clarity for their experimental design. However, some issues remain and are listed below.
1. “Traditional” teaching approach is mentioned throughout the paper (Lines 23, 99, 101, 174, 186, 196) but is not clarified until line 196 as “teacher-centered”. A clarification/explanation of “traditional” teaching approaches needs to be introduced at the onset of the paper.
2. Lines 43-49 discuss different anatomical factors involved in MSFA and the prevalence of two of the factors, but not the third (maxillary sinus walls). Also, the name of one of the anatomical factors changes from one line to the next (maxillary septum to septa).
3. The grammatical wording in the following lines contain misused words, spelling errors, incorrect tenses or awkward phrasing: Lines 35, 49, 58,69, 71, 83, 84, 122, 152, 156, 157, 167, 185, 228

Experimental design

4. Line 75 asserts “Such problems are best rooted in cases”. What is the basis of this assertion? Are their citations to back this claim up?
5. Line 106 states that the participants in the control group had “no discussion in or beyond each session” suggesting that the students had no interaction outside of class. I believe that the authors mean that there was no scheduled discussion time inside or outside of class, but I feel that this sentence asserts something different.
6. Who were the people involved in grading the exams? Were the exams graded blindly, or were the graders aware of which group was being graded?

Validity of the findings

7. Line 184 states that that this study was the first to apply a teaching method integrating PBL and CBL in a specific curriculum. This is inaccurate as other disciplines have combined both methodologies for teaching. For example, the citation below which is not included in the review article references.

Dong, Junhong & Zeng, Ping. (2017). The Application of CBL Teaching Combined with PBL Teaching Method in Biochemistry Experiment Teaching. 10.2991/meici-17.2017.126.

8. The discussion of study limitations beginning Line 225 to 228 reads like a bullet list of things that could have been done differently in the study, and should be expanded to involve an actual discussion of the study limitations and the ramifications therein.
9. Along with the above citation, it appears that the references are not as exhaustive of the available literature on PBL and CBL used in dental education as they could be. The following citation is an example of a pertinent publication that may further inform the current study.

An Overview of Case-Based and Problem-Based Learning Methodologies for Dental Education Nader A. Nadershahi, Daniel J. Bender, Lynn Beck, Cindy Lyon, Alexander Blaseio
Journal of Dental Education Oct 2013, 77 (10) 1300-1305

---

## Round 0.4 · Major Revisions

Your manuscript is much improved, but it still requires some important revisions. Please address all of the concerns of the reviewers. Most importantly, that they include these data from the questionnaire for the experimental group and the control group. If you cannot include data from both groups you should omit the questionnaire.

·

Basic reporting

Manuscript still needs editing for English language as there still remain some typos and grammatical errors.

Experimental design

Please include the results of the questionnaire for the control group. Currently, you only indicate satisfaction for the experimental group but do not provide the data for the control group even though both groups were provided with the questionnaire. A table would be appropriate so that readers could compare differences between experimental and control group. If no statistical analyses were conducted on these data please explain why. If you do not have data to report for both groups please omit the discussion and results of the questionnaire for the experimental group.

Validity of the findings

see notes on design.

·

Basic reporting

This article has greatly improved, and the authors have made most of the necessary changes to present their research clearly and concisely. However, some awkward sentence construction and word usage remain, and a few additions to the text are still needed for clarity. I am attaching an annotated pdf that contains my comments and suggestions.

Experimental design

no comment

Validity of the findings

no comment

---

## Round 0.5 · accepted · Accept

I believe that you have addressed the concerns of the reviewers.